# Naturally acquired antibodies against 7 *Streptococcus pneumoniae* serotypes in Indigenous and non-Indigenous adults

**Gabrielle N. Gaultier, Eli B. Nix, Joelle Thorgrimson, Douglas Boreham, William McCready, Marina Ulanova** [ID] *

NOSM University, Thunder Bay, ON, Canada

* mulanova@nosm.ca

**Data Availability Statement:** All relevant data are within the paper and its Supporting Information files.

## Abstract

Despite the use of pneumococcal conjugate vaccines for pediatric immunization, North American Indigenous populations continue to experience high burden of pneumococcal infections. Naturally acquired antibodies, which can protect unvaccinated adults against pneumococcal infections, have not previously been studied in Canadian Indigenous people. We analysed concentrations of natural serum IgG, IgM and IgA antibodies specific to 7 serotype-specific capsular polysaccharides (3, 6B, 9V, 14, 19A, 19F and 23F) in 141 healthy individuals (age between 18 and 80 years), including Indigenous adults living in 2 geographical different areas of Ontario, Canada, and non-Indigenous residing in northwestern Ontario. Regardless of the geographical area, concentrations of IgG specific to serotypes 6B, 9V, and 14, IgM specific to 9V, and all serotype-specific IgA were significantly higher in Indigenous study participants as compared to non-Indigenous. The differences are likely attributed to an increased exposure of Indigenous individuals to *Streptococcus pneumoniae* and/or cross-reactive antigens of other microorganisms or plants present in the environment. Although in non-Indigenous adults concentrations of IgM specific to 9V, 19A, 19F, and 23F significantly decreased with age, this was not observed in Indigenous individuals suggesting that Indigenous people may experience continuous exposure to pneumococci and cross-reactive antigens over the life span. Women had generally higher concentrations of natural IgG and IgM concentrations than men, with more striking differences found in Indigenous adults, potentially associated with larger exposure of women to young children, the major reservoir of pneumococci in communities. Our data suggest that increased rates of pneumococcal infections among Indigenous people are unlikely related to deficiency of naturally acquired antibodies, at least those specific to 7 common serotypes. Determining serological correlates of protection for adults will be essential to identify the groups in need of adult pneumococcal immunizations that may prevent excessive burden of the disease among North American Indigenous people.

**Funding:** This work received support from: Mitacs Accelerate with contribution by Bruce Power; IT05441; www.mitacs.ca/en/programs/accelerate to MU, DB; Northern Ontario Academic Medicine Association (AHSC AFP Innovation Fund); www.noama.ca; A-12-06 to WM, MU; Pfizer (Investigator Initiated Research Project, Grant 53232197) to M.U. https://www.cybergrants.com/pfizer/Research; and NOSM Summer Medical Student Research Awards to EBN and JT, https://www.nosm.ca/research/student-research-at-nosm-2/deans-summer-medical-student-research-awards/. The funders had no role in study design, data collection and analysis, decision to publish, or preparation of the manuscript.

**Competing interests:** Marina Ulanova received research funding from Pfizer through a grant for Investigator Initiated Research Project to her institution, honoraria for serving on the advisory boards, and travel expenses from Pfizer. This does not alter our adherence to PLOS ONE policies on sharing data and materials. All other authors have declared that no competing interests exist.

## Introduction

*Streptococcus pneumoniae* (pneumococcus) is Gram-positive, encapsulated, extracellular bacterium, which asymptomatically colonizes mucosal surfaces of the upper respiratory tract and spreads through respiratory droplets [1,2]. When pneumococci evade the host defenses, they can cause non-invasive (otitis media, sinusitis, pneumonia) and invasive (meningitis, septicaemia, pericarditis) infections, if the bacteria invade normally sterile body sites [3,4]. Children under 5 years of age, the elderly, and immunocompromised adults have the highest incidence of invasive pneumococcal disease (IPD) and pneumococcal pneumonia [2,3]. Over 90 pneumococcal serotypes have been identified through antigenic differences in their polysaccharide capsules [5]. The capsule is a key virulence factor in the pathogenesis of pneumococcal disease; it facilitates colonization of mucosal surfaces, inhibits the classic and alternative complement pathways, and reduces opsonization by impeding interactions between complement fragments or antibodies with their receptors on phagocytes [1,6].

Since the introduction of pneumococcal conjugate vaccines (PCV), which stimulate T-cell dependent production of antibodies to pneumococcal capsular polysaccharides, into pediatric immunization programs two decades ago, incidence rates of IPD caused by vaccine serotypes decreased among both immunized children and unimmunized adults. The indirect effect of pediatric immunization is the result of decreased transmission of *S. pneumoniae* from immunized children to adults due to reduced nasopharyngeal colonization [7]. In the post-PCV era, the colonization rates of vaccine serotypes have also decreased in adults [8,9]. In Canada, PCV containing 7 serotype-specific pneumococcal capsular polysaccharides, *i.e.* 4, 6B, 9V, 14, 18C, 19F, and 23F (PCV7) introduced into publicly-funded universal pediatric immunization program in 2005, was replaced by PCV13, which includes 6 additional serotypes (1, 3, 5, 6A, 7F, 19A), in 2010–2011 [10]. The 23-valent pneumococcal polysaccharide vaccine (PPV23) containing purified capsular polysaccharides of serotypes 1, 2, 3, 4, 5, 6B, 7F, 8, 9N, 9V, 10A, 11A, 12F, 14, 15B, 17F, 18C, 19F, 19A, 20, 22F, 23F, and 33F is currently used for immunization of adults $\geq$65 years of age as well as groups of risk for IPD >2 years of age [11]. However, pneumococcal vaccine uptake in Canadian adults is low; amongst the adult population with chronic conditions predisposing to IPD, only 17.3% of 18–64 year-old and 36.5% of $\geq$65 year-old individuals received PPV23 in 2014 [12]. Similar to other countries, in Canada, over the last years, the incidence of IPD caused by PCV13 serotypes has decreased in both young children and older adults although the overall rates did not decrease mainly due to the increasing incidence of PPV23 unique and non-vaccine serotypes [13]. While the numbers of hospitalized adult cases of community-acquired pneumococcal pneumonia decreased between 2011 and 2014, they increased in 2015, with a particular growth in cases caused by serotype 3 [10].

The province of Ontario is a large region of over one million square kilometers characterized by diverse climate conditions, ranging from subarctic in the north to moderate humid continental climate in the south. Northwestern Ontario has the largest proportion of Indigenous people, i.e. 25.4% of the population compared to 2.83% in the whole province [14,15]. In North America, the incidence of IPD remains higher in Indigenous people compared to the general population despite the use of pneumococcal vaccines [16–19]. Eton *et al.* (2017) investigated the cases of IPD admitted to a hospital serving Indigenous communities in northwestern Ontario. The incidence of IPD over a period of 2010–2015 was more than double the 2013 rate for all of Canada (23.1 vs. 9.0/100,000/year) [20]. Among IPD cases admitted to the major hospital serving northwestern Ontario between 2006 and 2015, 29.1% of adult patients were Indigenous, and 72% of cases were caused by non-PCV13 serotypes [21].

We recently studied immune response to PCV13 in adults with severe chronic kidney disease in northwestern Ontario and unexpectedly found higher concentrations of naturally

acquired 6B, 9V, 14, 19F, and 23F serotype-specific IgG antibodies in Indigenous individuals as compared to their non-Indigenous counterpart [22]. As no studies have previously addressed natural pneumococcal antibodies in Canadian Indigenous people, we extended the analysis to include healthy adults residing in different parts of the province to account for potential effect of distinct environmental conditions. The analysis involved serum IgG, IgM and IgA antibodies to 7 serotype-specific capsular polysaccharides included into PCV13 and PPV23 (3, 6B, 9V, 14, 19A, 19F and 23F). We found that regardless of the geographical area, concentrations of several serotype-specific IgG and IgM, and all serotype-specific IgA were higher in Indigenous study participants as compared to non-Indigenous individuals residing in northwestern Ontario.

## Materials and methods

### Ethics statement

The study has been conducted according to the World Medical Association Declaration of Helsinki. We adhered to the principles of Ownership, Control, Access, and Possession (OCAP) as defined by the National Aboriginal Health Organization [23], and the guidelines of Canadian Tri-Council Policy Statement: Ethical Conduct for Research Involving Humans (TCPS2), specifically those outlined in Chapter 9: Research Involving First Nations, Inuit and Métis Peoples of Canada [24]. The study was approved by the research ethics board of Lakehead University (REB #117 11-12/Romeo #1462496, Thunder Bay, Ontario).

### Study participants

The study included 141 healthy adults (Table 1). Indigenous adults were recruited from two Ojibwa First Nations communities, one in southern Ontario (Group 1) and the other in northwestern Ontario (Group 2). Non-Indigenous adults were recruited from two cities in

**Table 1. Participant demographics.**

| Group | Number | Age (years) | | | % Female |
|---|---|---|---|---|---|
| | | Mean ± SEM | Median | Range | |
| Indigenous [a] | 77 | 41 (1.67) | 39 | 18–80 | 49.4 |
| non-Indigenous [b] | 64 | 45 (2.11) | 52 | 20–72 | 66.6* |
| Group 1 [c#] | 30 | 46 (2.75) ** | 51 | 20–80 | 50 |
| Group 2 [d] | 47 | 37 (1.94) | 34 | 18–67 | 48.9 |
| Group 3 [e] | 45 | 46 (2.48) ** | 52 | 22–72 | 62.2 |
| Group 4 [f&] | 19 | 43 (4.04) | 48 | 20–68 | 77.8 |

[a] Indigenous: Indigenous First Nations adults residing in Ontario (combination of groups 1 and 2).

[b] non-Indigenous: Non-Indigenous adults residing in Ontario (combination of groups 3 and 4).

[c] Group 1: Indigenous adults residing in a southern Ontario First Nations Community.

[d] Group 2: Indigenous adults residing in a northwestern Ontario First Nations Community.

[e] Group 3: Non-Indigenous adults residing in Thunder Bay, Ontario.

[f] Group 4: Non-Indigenous adults residing in Kenora, Ontario.

[#] Age of one participant is unknown.

[&] Sex of one participant is unknown.

[*] $p < 0.05$, Indigenous females vs. non-Indigenous females (Fisher's exact test).

[**] $p < 0.01$, in comparison to group 2 (Mann-Whitney U test).

SEM, Standard error of the mean.

No significant differences in the proportions of females between groups 1–4 (Fisher's exact test).

northwestern Ontario, Thunder Bay (Group 3) and Kenora (Group 4). Participants were considered eligible if they were over the age of 18 years, self-declared generally healthy, did not have a history of taking immunosuppressive medications, and declared that they did not receive a pneumococcal vaccine. Ethnicity was determined based on self-declaration. Serum samples from both Indigenous communities, Kenora, and a part of the Thunder Bay samples were collected during January—November 2015, as we previously described [25]. The remaining Thunder Bay samples were collected between November 2016 and November 2017. The samples were collected under informed written consent and stored at -80˚C prior to analysis.

### Antibody analysis

Serum concentrations of IgG, IgM and IgA specific to the pneumococcal serotypes 3, 6B, 9V, 14, 19A, 19F and 23F were determined by enzyme-linked immunosorbent assay (ELISA) according to the WHO protocol [26].

### Statistical analysis

The serotype-specific antibodies were reported as geometric mean concentrations (GMC) with two-sided 95% confidence intervals (CI). Groups were compared using Mann-Whitney U test, or Kruskal-Wallis test with Dunn's ad hoc tests depending on the number of groups compared; Fishers' exact test was conducted to compare categorical variables. To study the association of antibody concentrations with age, linear regression and Spearman correlation analyses were performed. A $p$ value of $< 0.05$ was reported as statistically significant. Statistical analysis was performed using Graph-Pad Prism 9 (GraphPad Prism Software Inc., San Diego, CA).

## Results

Demographics of 141 study participants are presented in Table 1. The age range was between 18 and 80 years, without significant differences between Indigenous and non-Indigenous individuals. The non-Indigenous participants (Group 3 and Group 4 combined) had a higher proportion of females compared to the Indigenous (Group 1 and Group 2 combined). Group 2 was significantly younger compared to Groups 1 or 3 and it had the lowest proportion of women while Group 4 had the highest (Table 1).

Indigenous individuals (Group 1 and Group 2 combined) had significantly higher concentrations of 3 out of 7 serotype-specific IgG (anti-6B, -9V, and -14) compared to non-Indigenous (Fig 1). In addition, proportions of Indigenous adults who had concentrations of IgG specific to serotypes 6B, 9V, and 14 above 1.0 μg/mL and above 1.3 μg/mL were significantly higher than the corresponding proportions of non-Indigenous individuals (**S1 Table**). Among Indigenous individuals, GMC for IgG specific to serotypes 19A and 19F were significantly higher in Group 1 than Group 2; no differences in GMC between non-Indigenous individuals residing in different communities (Group 3 vs. Group 4) were present (Table 2A). Concentrations of serotype-specific IgM were generally similar among all the groups (Table 2B) although Indigenous individuals (Group 1 and Group 2 combined) had higher IgM specific to 9V compared to non-Indigenous (Group 3 and Group 4 combined), as demonstrated in Fig 2. In contrast, concentrations of all 7 serotype-specific IgA were significantly higher in Indigenous than non-Indigenous study participants (Fig 3). Although no differences were present between Indigenous (between Groups 1 and 2) or non-Indigenous individuals (between Groups 3 and 4) residing in different communities, GMC of specific IgA in Indigenous people living in southern Ontario (Group 1) were significantly higher for serotypes 9V, 14, and 23F as compared to Group 3, and for serotypes 9V, 14, and 19A compared to Group 4. Similarly,

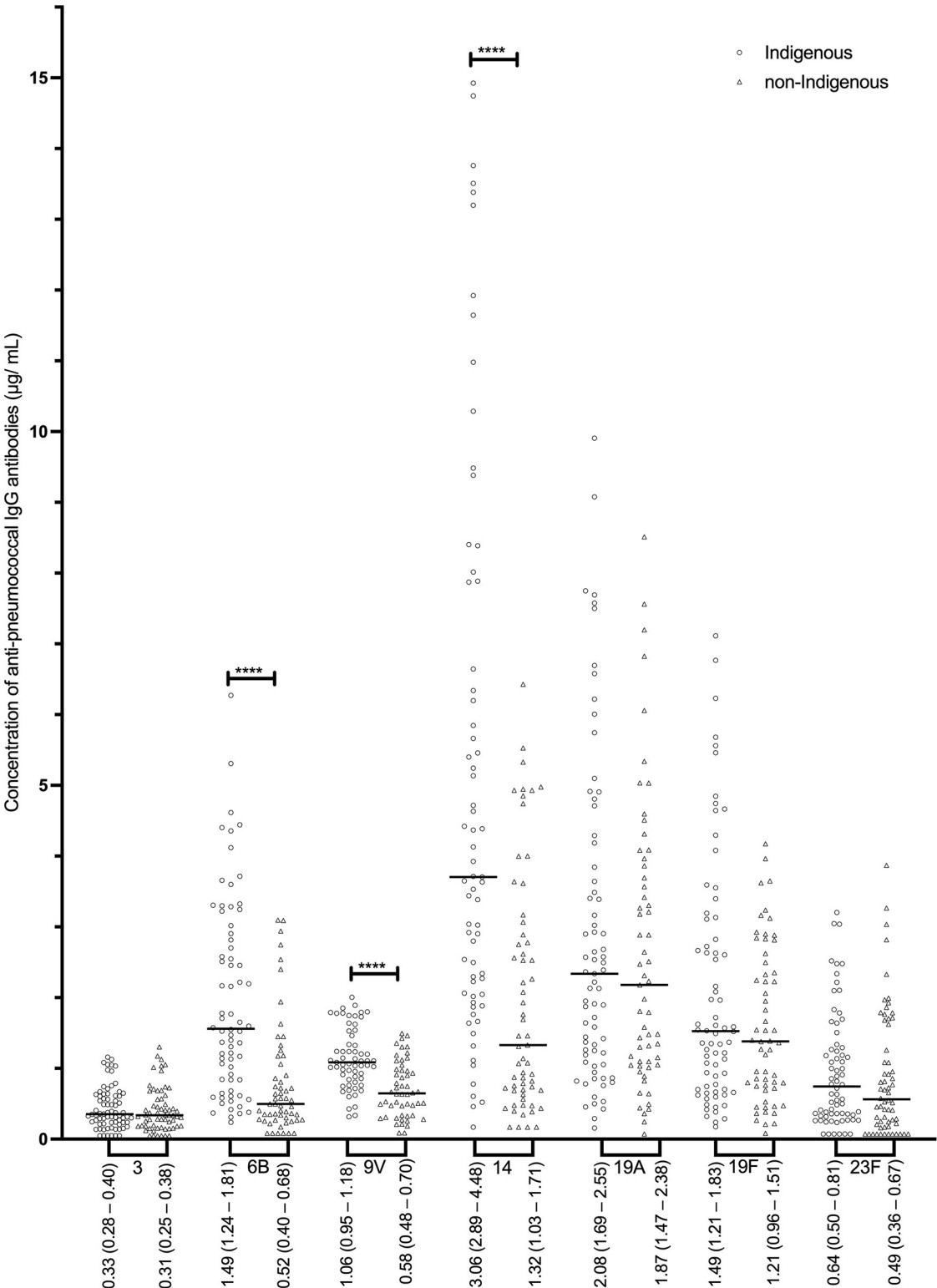

**Fig 1. Concentrations of serotype-specific pneumococcal IgG antibodies for Indigenous (n = 77) and non-Indigenous groups (n = 64).** Geometric mean antibody concentrations (GMC) are displayed (μg/ mL) and the 95% confidence intervals of the GMC. Statistical significance was determined using a Mann-Whitney U test for all serotypes. **** p < 0.0001.

**Table 2A. The geometric mean concentrations of anti-pneumococcal IgG antibodies (μg/ mL, 95% confidence intervals of the geometric mean).**

| Serotype | Group 1 | Group 2 | Group 3 | Group 4 | p value (Kruskal-Wallis test) |
|---|---|---|---|---|---|
| 3 | 0.39 (0.29–0.53) | 0.29 (0.23–0.37) | 0.25 (0.19–0.33) | 0.45 (0.35–0.57) | 0.019 |
| 6B | 1.59 (1.13–2.23) | 1.47 (1.17–1.86) $$$$ | 0.46 (0.34–0.61) #### | 0.68 (0.38–1.22) ^ & | <0.0001 |
| 9V | 1.02 (0.86–1.21) | 1.09 (0.94–1.25) $$$$ | 0.53 (0.41–0.67) ## | 0.73 (0.55–0.95) & | <0.0001 |
| 14 | 4.50 (3.02–6.72) | 3.18 (2.44–4.15) $$$ | 1.28 (0.93–1.77) #### | 1.43 (0.94–2.18) ^^ & | <0.0001 |
| 19A | 2.98 (2.03–4.39) * | 1.60 (1.28–1.99) | 1.65 (1.20–2.27) | 2.48 (1.77–3.49) | 0.0088 |
| 19F | 2.33 (1.57–3.45) ** | 1.20 (0.95–1.51) | 1.08 (0.80–1.46) ## | 1.53 (1.10–2.12) | 0.0043 |
| 23F | 0.88 (0.59–1.32) | 0.54 (0.40–0.72) | 0.38 (0.26–0.56) # | 0.84 (0.59–1.19) | 0.013 |

Group 1: Indigenous adults residing in a southern Ontario First Nations Community.

Group 2: Indigenous adults residing in a northwestern Ontario First Nations Community.

Group 3: Non-Indigenous adults residing in Thunder Bay, Ontario.

Group 4: Non-Indigenous adults residing in Kenora, Ontario.

Significance determined by Dunn's multiple comparisons test.

Group 1 vs. Group 2

* $p < 0.05$

** $p < 0.01$.

Group 1 vs. Group 3

# $p < 0.05$

## $p < 0.01$

#### $p < 0.0001$.

Group 1 vs. Group 4

^ $p < 0.05$

^^ $p < 0.01$.

Group 2 vs. Group 3

$$$ $p < 0.001$

$$$$ $p < 0.0001$.

Group 2 vs. Group 4

& $< 0.05$.

Group 3 vs. Group 4: No significant differences for any serotypes.

**Table 2B The geometric mean concentrations of anti-pneumococcal IgM antibodies (μg/ mL, 95% confidence intervals of the geometric mean).**

| Serotype | Group 1 | Group 2 | Group 3 | Group 4 | p value (Kruskal-Wallis test) |
|---|---|---|---|---|---|
| 3 | 0.28 (0.21–0.37) | 0.27 (0.22–0.34) | 0.28 (0.23–0.34) | 0.20 (0.16–0.26) | > 0.05 |
| 6B | 0.98 (0.72–1.34) | 1.11 (0.89–1.37) | 1.01 (0.81–1.27) | 0.84 (0.61–1.17) | > 0.05 |
| 9V | 0.34 (0.24–0.49) | 0.41 (0.33–0.50) | 0.26 (0.20–0.34) | 0.23 (0.15–0.37) | > 0.05 |
| 14 | 0.48 (0.35–0.66) | 0.61 (0.51–0.75) | 0.55 (0.44–0.68) | 0.38 (0.28–0.53) | > 0.05 |
| 19A | 1.31 (0.97–1.76) | 1.27 (1.03–1.57) | 1.27 (1.01–1.60) | 1.02 (0.73–1.43) | > 0.05 |
| 19F | 1.27 (0.93–1.73) | 1.20 (1.01–1.43) | 1.39 (1.11–1.74) | 1.24 (0.91–1.69) | > 0.05 |
| 23F | 0.22 (0.16–0.28) | 0.24 (0.19–0.29) | 0.21 (0.17–0.25) | 0.16 (0.12–0.22) | > 0.05 |

Group 1: Indigenous adults residing in a southern Ontario First Nations Community.

Group 2: Indigenous adults residing in a northwestern Ontario First Nations Community.

Group 3: Non-Indigenous adults residing in Thunder Bay, Ontario.

Group 4: Non-Indigenous adults residing in Kenora, Ontario.

Dunn's multiple comparisons test did not detect any statistically significant differences in IgM concentrations between groups for any serotype.

**Table 2C The geometric mean concentrations of anti-pneumococcal IgA antibodies (μg/ mL, 95% confidence intervals of the geometric mean).**

| Serotype | Group 1 | Group 2 | Group 3 | Group 4 | p value (Kruskal-Wallis test) |
|---|---|---|---|---|---|
| 3 | 0.20 (0.14–0.30) | 0.24 (0.18–0.32) | 0.16 (0.13–0.21) | 0.17 (0.11–0.25) | > 0.05 |
| 6B | 0.12 (0.09–0.17) | 0.13 (0.10–0.15) | 0.09 (0.07–0.11) | 0.09 (0.06–0.13) | > 0.05 |
| 9V | 0.17 (0.13–0.24) | 0.21 (0.18–0.26) $$$$ | 0.10 (0.08–0.12) ## | 0.07 (0.05–0.11) ^^ &&&& | <0.0001 |

(*Continued*)

**Table 2A.** (Continued)

| 14 | 0.32 (0.22–0.46) | 0.21 (0.16–0.28) | 0.14 (0.11–0.18) ## | 0.12 (0.07–0.21) ^^ | 0.0008 |
|---|---|---|---|---|---|
| 19A | 0.26 (0.19–0.37) | 0.27 (0.21–0.34) $ | 0.16 (0.14–0.19) | 0.14 (0.10–0.18) ^ & | 0.0013 |
| 19F | 0.11 (0.08–0.16) | 0.10 (0.08–0.12) | 0.08 (0.07–0.10) | 0.07 (0.06–0.10) | > 0.05 |
| 23F | 0.11 (0.07–0.16) | 0.11 (0.09–0.14) $ $ | 0.06 (0.04–0.07) # | 0.07 (0.04–0.10) | 0.0017 |

Group 1: Indigenous adults residing in a southern Ontario First Nations Community.

Group 2: Indigenous adults residing in a northwestern Ontario First Nations Community.

Group 3: Non-Indigenous adults residing in Thunder Bay, Ontario.

Group 4: Non-Indigenous adults residing in Kenora, Ontario.

Significance determined by Dunn's multiple comparisons test.

Group 1 vs. Group 2: No significant differences for any serotypes.

Group 1 vs. Group 3

# p < 0.05

## p < 0.01.

Group 1 vs. Group 4

^ p < 0.05

^^ p < 0.01.

Group 2 vs. Group 3

$ p < 0.05

$ $ p < 0.01

$ $ $ $ p < 0.0001.

Group 2 vs. Group 4

& p < 0.05

&&&& p < 0.0001.

Group 3 vs. Group 4: No significant differences for any serotypes.

Indigenous people living in northwestern Ontario (Group 2) had higher GMC of IgA specific to 9V, 19A, and 23F as compared to Group 3, and to 9V and 19A compared to Group 4 (Table 2C).

To determine if concentrations of natural pneumococcal antibodies were dependent on age, linear regression analyses were performed for Indigenous and non-Indigenous groups for each of the seven serotype-specific IgG, IgM and IgA. No consistent or clear association of most IgG (S2 Table) or IgA (S3 Table) concentrations with age for either group was observed. However, concentrations of IgM specific to serotypes 9V, 19A, 19F, and 23F significantly decreased with age in non-Indigenous (Fig 4A), but not in Indigenous individuals (Fig 4B). Spearman correlation analysis indicated statistically significant, but weak negative correlation between age of non-Indigenous adults and concentrations of IgM specific to serotypes 9V (r = −0.28, p < 0.05) and 19F (r = −0.25, p < 0.05). No significant correlation between age and the rest of serotype-specific IgM concentrations was found (S4 Table).

As differences in antibody concentrations between men and women could potentially account for the above observations, we compared GMC of serotype-specific IgG, IgM, and IgA between male and female individuals of the same ethnic group. No significant differences in mean ages between males and females of the same ethnicity were present (S5 Table). As shown in S6 Table, concentrations of IgG and IgM antibodies were generally higher in Indigenous women than men, with statistically significant differences identified for 14-, 19A-, and 19F-specific IgG, and for all but one serotype-specific IgM (except for serotype 14). In non-Indigenous individuals, higher female IgG antibody concentrations were found only in case of serotype 23F; all but one serotype-specific IgM (except for serotype 6B) were significantly higher in

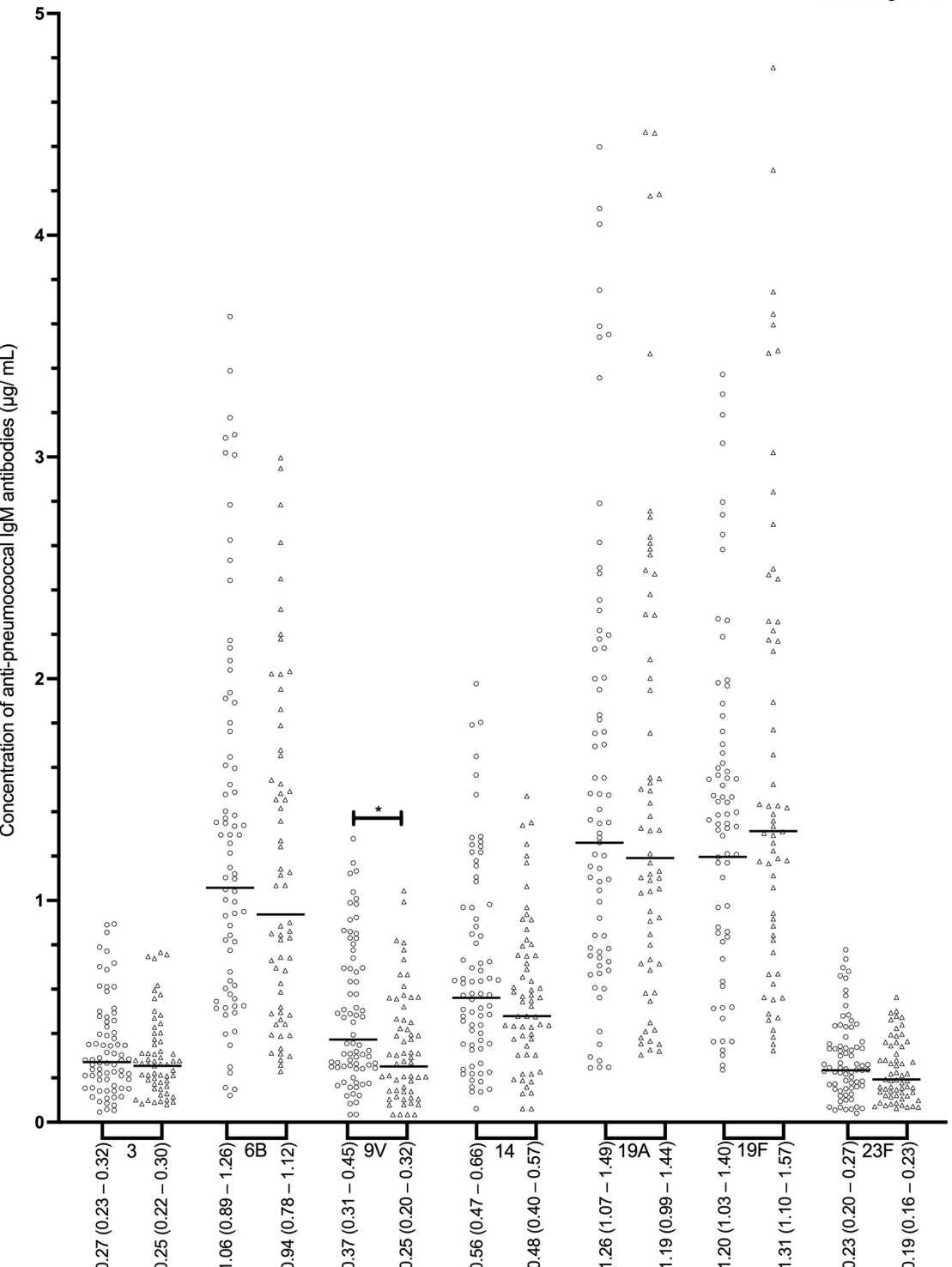

**Fig 2. Concentrations of serotype-specific pneumococcal IgM antibodies for Indigenous (n = 77) and non-Indigenous groups (n = 64).** Geometric mean antibody concentrations (GMC) are displayed (μg/ mL) and the 95% confidence intervals of the GMC. Statistical significance was determined using a Mann-Whitney U test for all serotypes. * p < 0.05.

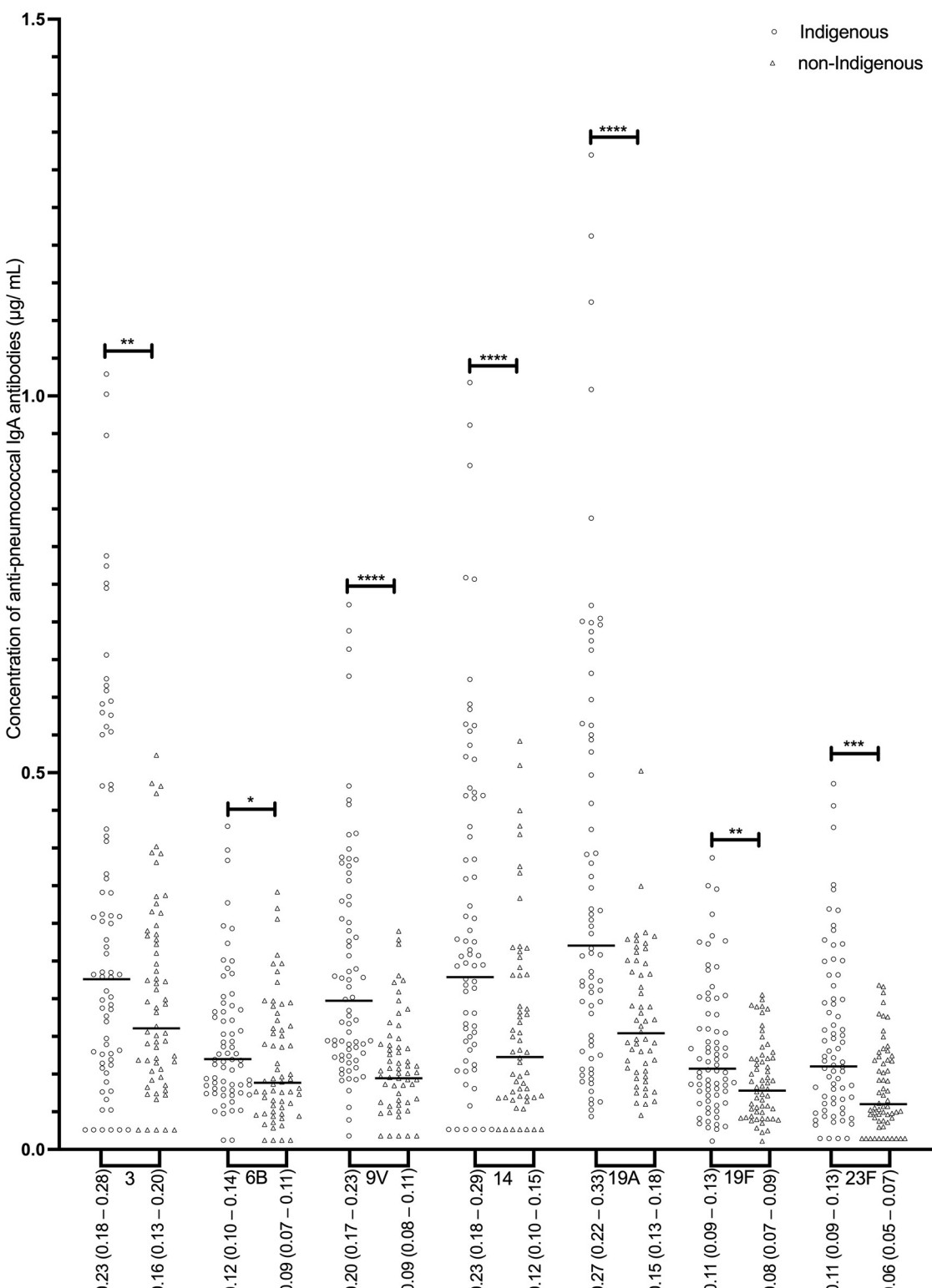

**Fig 3. Concentrations of serotype-specific pneumococcal IgA antibodies for Indigenous (n = 77) and non-Indigenous groups (n = 64).** Geometric mean antibody concentrations are displayed (µg/ mL) and the 95% confidence intervals of the GMC. Statistical significance was determined using a Mann-Whitney U test for all serotypes. $^*$ p < 0.05, $^{**}$ p < 0.01, $^{***}$ p < 0.001, $^{****}$ p < 0.0001.

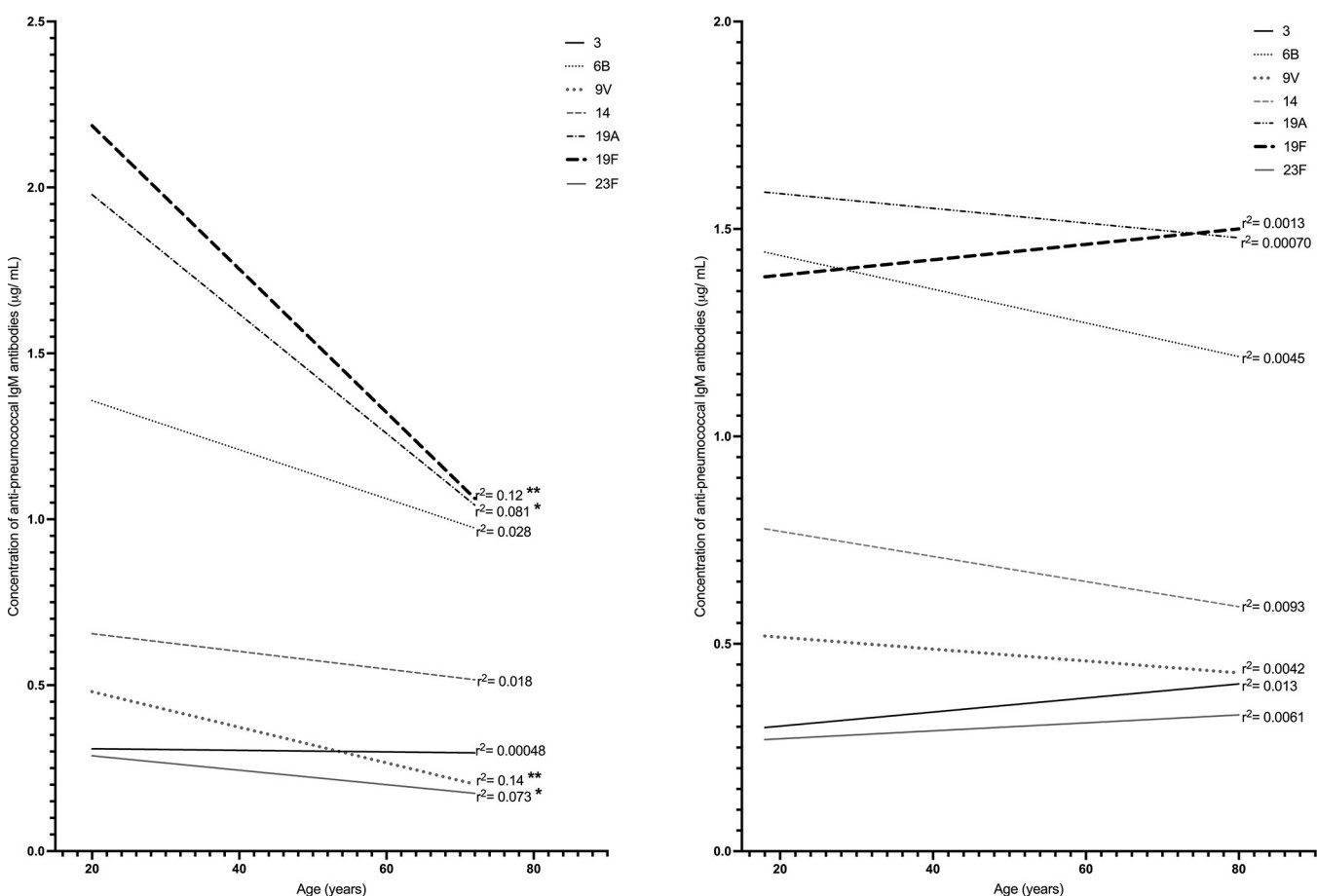

**Fig 4.** Linear regression analysis of association of serotype-specific IgM antibodies with age for (A) non-Indigenous groups (n = 64) and (B) Indigenous (n = 76). The goodness of the fit is reported by the $r^2$ value; the p value determines if the slope is significantly non-zero. * $p < 0.05$, ** $p < 0.01$.

female than male individuals. Concentrations of IgA antibodies did not differ between male and female individuals of either ethnicity (S6 Table).

## Discussion

In this study, we quantified natural antibodies specific to seven pneumococcal serotypes (3, 6B, 9V, 14, 19A, 19F and 23F), which are all included in currently used pneumococcal vaccines (PPV23 and PCV13) owing to their epidemiological importance. These serotypes have distinct biological characteristics and different prevalence in various forms of pneumococcal infections [27].

The principal rationale for specifically selecting these 7 pneumococcal serotypes was based on their distinct immunogenic characteristics recognized from studies of antibody response to multi-component pneumococcal vaccines. It is known that serological thresholds for protection against infections caused by these serotypes substantially differ. A large multi-national study by Voysey et al. (2018) demonstrated that antibody concentrations associated with protection against carriage greatly varied between 10 different serotypes (1, 4, 5, 6B, 7F, 9V, 14, 18C, 19F, 23F). The highest protective threshold was found for serotypes 14 and 19F, and the lowest for serotypes 6B and 23F; the threshold for serotype 9V was intermediate [28]. Including analysis of antibody for serotype 3 was of special interest for our study. Pneumococci of

this serotype are heavily encapsulated are highly resistant to phagocytosis. As serotype 3 is typically associated with invasive disease rather than with nasopharyngeal carriage, this may result in poor development of natural antibody [29]. We were interested in antibodies to serotype 14 because its capsule is structurally similar to the capsular polysaccharide of Group B *Streptococcus* type III that could influence the development of natural immunity; immunological cross-reactivity between these bacteria was reported [30]. Considering that pneumococci of the serogroup 19 widely cross-react with other common bacteria including *Klebsiella* and streptococci, we were interested in the analysis of naturally-acquired antibodies against serotypes 19A and 19F [31]. In addition, antibiotic-resistant serotype 19A emerged as replacement disease following the introduction of the first-generation pneumococcal conjugate vaccine (PCV7) in early 2000s, and became highly significant in the epidemiology of pneumococcal infections until immunization with PCV13 greatly reduced its prevalence [32].

Prior to the introduction of pneumococcal conjugate vaccines, all these 7 serotypes were highly significant in epidemiology of IPD as well as commonly associated with nasopharyngeal carriage in young children [33,34]. In Canada, the rates of disease caused by PCV13 vaccine serotypes (except for serotype 3), including IPD in all age groups, and pneumococcal pneumonia in adults, have greatly decreased since the implementation of PCV13 [13]. According to the most recent data by the National Laboratory Surveillance of Invasive Streptococcal Disease (2017), among all IPD cases, the prevalence of disease caused by serotypes 6B, 9V, 14 or 23F was less than 1%, 2.5% for 19F and 4.8% for 19A, while serotype 3 accounted for 9.7% [35]. We observed a decrease in the prevalence of PCV13 serotypes in the etiology of adult IPD in northwestern Ontario indicating an indirect effect of pediatric immunization in a region with a large population of Indigenous people [21].

Historically, North American Indigenous people experienced the highest IPD incidence rates in the world [36]. Pre-PCV-era population-based studies conducted in the White Mountain Apache reported the highest global incidence rates of IPD, i.e. 2396 cases per 100,000 population of children between 1 and 2 years of age [37]. In Alaska, annual incidence of IPD among Indigenous children < 2 years of age was 624 per 100,000, with meningitis or pneumonia rates being 8–10 times higher than elsewhere in the U.S. [38]. Although the rates of IPD greatly decreased in Indigenous children following the introduction of PCV, they remained over 5-fold higher in White Mountain Apache children compared to the same age group in the general U.S. population [39]. Epidemiological studies identified several modifiable risk factors for IPD in Indigenous adults, including renal failure, alcoholism, and unemployment that could potentially explain a large difference in incidence rates between the Indigenous and the general U.S. population [40]. While the introduction of PCV led to elimination of IPD due to vaccine serotypes, incidence rates of disease caused by non-vaccine serotypes increased in Alaska [41]. That study identified the lack of in-home piped water as an essential factor associated with high incidence rates of IPD in Alaska. Similarly, increased incidence of IPD in Indigenous people as compared to the general population has been reported from Canada including our region of interest, northwestern Ontario [18–21]. Nevertheless, despite a decreasing prevalence of PCV13 serotypes in the etiology of IPD in northwestern Ontario, the proportion of PCV or non-vaccine serotypes did not differ between Indigenous and non-Indigenous adults with IPD [21].

What can be the reason/s for the differences in concentrations of naturally acquired pneumococcal antibodies between Indigenous and non-Indigenous study participants? Indeed, healthy adult Indigenous residents of both Ojibwe communities, one in northwestern, another in southern Ontario, had significantly higher concentrations of three serotype-specific IgG, one serotype-specific IgM and all seven serotype-specific IgA as compared to their non-Indigenous counterparts in northwestern Ontario. Of note, the two communities have considerable

differences in geographical location, climate, environment, distance to large urban centers, and access to services. One in the north is remote and exposed to harsh weather during winters. The southern one has a short-distance access to a large city, more developed services on-site, and more favourable climate with milder winters and warm summers. Regardless of these differences, people in both Indigenous communities have stronger natural pneumococcal immunity as compared to non-Indigenous healthy adults living in a smaller town of Kenora (15,000 habitants), which is only 80 km away from the northwestern Indigenous community, or in a larger northwestern city of Thunder Bay (110,000 habitants).

One possibility could be higher carriage rates in Indigenous compared to non-Indigenous communities, as more frequent exposure to pneumococci would contribute to higher concentrations of natural pneumococcal antibodies. The nasopharyngeal colonization and repeated exposure to pneumococci are important for immune defenses in adults; these are immunizing events that result in the production of natural antibodies and maintain the memory B cell pool [42]. Even in the setting of high PCV immunization rates, substantial burden of pneumococcal colonization persisted in children and adults in North American Indigenous communities, with children being the major drivers of intra-household transmission [43,44]. According to our data, women had generally higher concentrations of natural IgG and IgM antibodies than men, with more striking differences found in Indigenous groups, that is in line with earlier observations in a different population [45]. Among 30–64 year old Finnish people, concentrations of naturally acquired pneumococcal IgG and IgM antibodies were higher in women than men, with significant differences identified for 4/6 serotype-specific IgG (3, 6B, 14, and 23F) and all serotype-specific IgM (3, 4, 6B, 9V, 14, and 23F) that could be related to larger exposure of women to children [45]. Indeed, studies conducted in various populations in pre- and post-PCV time demonstrated that pneumococcal carriage in adults is directly associated with exposure to young children [8,46].

It is plausible to suggest that acquisition of natural pneumococcal antibodies by adults may be caused by persistent carriage of *S. pneumoniae* in children within Indigenous communities. Although the latter may potentially be related to insufficient pediatric immunization coverage in the region, no data to support or rule out this idea are available. No publicly accessible data are available on immunization coverage in specific communities of Ontario, while the overall PCV13 vaccination coverage of children by two years of age in Canada is 81.4% [47]. Based on official data by Public Health Ontario, the average rates of immunization of school pupils in Ontario with pneumococcal conjugate vaccine during the time of our study were the following: 2015–2016: 79.0%, 2016–2017: 79.7%, 2017–2018: 74.1%; however, immunization coverage reports in this province do not include any information of the race or ethnicity [48]. For comparison, in the remote Nordic region of Nunavik (province of Québec), with 90% of the population being Indigenous, the uptake rates of PCV7 in 2005 were similar to those in the whole province of Québec, and higher than the average Canadian data [49]. However, despite high immunization rates, pneumococcal disease burden in Nunavik Indigenous children remained higher that elsewhere in Canada [50]. Overall, Indigenous people in Canada continue to experience poorer health and higher mortality rates compared to the general population, attributed to the consequences of colonialism and unfavourable social determinants of health such as poverty, overcrowded housing, and inadequate access to clean water [51]. In addition to high rates of IPD, significant burden of other invasive bacterial diseases, including *H. influenzae*, group A and B streptococcal infections, and meningococcal disease has been reported in Indigenous communities [52].

However, persistent carriage of pneumococci in children is not exclusive in Indigenous people and may depend on specific characteristics of individual serotypes and their interactions with other bacteria sharing the nasopharyngeal microbiome. Different immunogenicity

of pneumococcal serotypes can influence their abilities to colonize the nasopharynx; notably, capsular polysaccharides of serotypes 6B, 14, and 19F are poorly immunogenic [53,54]. Salt *et al.* (2007) suggested that the low immunogenicity of serotype 6B explained its propensity to frequently colonize the nasopharynx of young children [55]. In addition to dissimilar immunogenic properties, pneumococcal serotypes have different virulence; some are more commonly associated with invasive disease rather than carriage [29]. One example is serotype 3, which has a thick polysaccharide capsule and is highly resistant to phagocytosis resulting in the most severe forms of pneumococcal infection [29]. In our study, both Indigenous and non-indigenous individuals had the lowest concentrations of antibody against serotype 3 among all the studied serotype-specific IgG, suggesting poor natural immunity against this serotype. Notably, current pediatric PCV13 immunization decreased the prevalence of all vaccine serotypes in etiology of pneumococcal infections, with exception of IPD caused by serotype 3 [10]. Although the serological correlates of protection against adult IPD are unknown, studies in children indicated higher immunological threshold for serotype 3 than for other PCV13 serotypes [28,56].

Natural antibodies to pneumococcal capsular polysaccharides can be induced by the exposure to cross-reactive microbial or environmental antigens. In particular, common oral non-pathogenic streptococci, *i.e. S. mitis*, *S. orali*, and *S. infantis* are able to carry capsular polysaccharides cross-reactive to *S. pneumoniae* and may then contribute to acquisition of natural pneumococcal antibodies [57]. As prevalence, distribution, and exposure to cross-reactive antigens depends on the environment, lifestyle, and hygienic practices, their role in the development of natural immunity may be more prominent in Indigenous than non-Indigenous communities. Earlier studies demonstrated serological cross-reactivity between capsular polysaccharides of pneumococcal serotype 6B and *Haemophilus influenzae* serotype a (Hia) [58]. Considering that Hia is a pathogen which is more prevalent among North American Indigenous people as compared to the general population [59], and invasive Hia disease has consistently been present in northwestern Ontario over the last 2 decades [60,61], cross-reactive antibody induced by exposure to Hia may contribute to the development of natural antibody specific to serotype 6B among Indigenous adults. Indigenous people living in rural communities may also be frequently exposed to cross-reactive antigens of *Klebsiella pneumoniae*, *Pseudomonas aeruginosa*, or certain fungi common in the natural environment [62], as well as some derived from plant polysaccharides traditionally consumed by Indigenous people for nutrition or medicine, such as fruit pectin or *Althaea officinalis* [63,64].

Published data on natural pneumococcal antibody in North American Indigenous adults are limited by 2 studies of several serotype-specific IgG conducted in a different population, elderly Alaska Native people, precluding direct comparison to our data [65,66]. It is even more difficult to compare our data to published seroprevalence data collected in other adult populations (U.S., UK, Finland, The Netherlands, Australia) that used different assay methods and reported a wide range of antibody concentrations. Nevertheless, the concentrations of specific IgG in our sample are similar to the previously reported data (S7 Table). Published studies mostly addressed IgG antibodies with only data available on concentrations of naturally acquired serotype-specific IgM in a Finnish population [45]. Although IgG is a dominant immunoglobulin, which plays essential role in opsonisation of pneumococci, IgM is characterized by superior abilities to activate complement, and IgA can trigger phagocytosis of pneumococci by neutrophils [67,68]. Despite the differences in populations, our data show similarities with data by Simell *et al.* 2008, *i.e.* higher concentrations of several serotype-specific IgM in women than men, and a decrease in concentrations of natural IgM antibody with age [45]. However, we observed significant age-dependent decrease in IgM specific to serotypes 9V, 19A, 19F, and 23F in non-Indigenous, rather than Indigenous study participants. These

observations imply that older Indigenous people may experience continuous exposure to cross-reactive antigens over their life span. To the best of our knowledge, no previous studies of natural pneumococcal IgA antibodies have been conducted. Although concentrations of serum IgA antibodies were inferior to IgG or IgM, the former were significantly higher in Indigenous than non-Indigenous adults that may potentially reflect differences in antigenic stimulation through the mucosal surfaces. While the role of secretory IgA in mucosal defences against bacterial infections has been well established, the function of serum IgA is less known. Earlier studies have demonstrated that capsular polysaccharide-specific serum IgA antibodies are able to opsonize pneumococci for phagocytosis by neutrophils via the process involving the alternative complement pathway and interaction with IgA Fc receptor (FcαRI) [68,69]. Recent data indicated that IgA was more efficient, than IgG, in the formation of neutrophil extracellular traps, suggesting yet another mechanism of anti-bacterial action of serum IgA [70,71]. Considering that serum concentrations of natural IgA antibodies are low, a question to what degree serum IgA may contribute to natural pneumococcal immunity deserves further study.

Considering higher burden of IPD in Indigenous than non-Indigenous people, our findings of higher concentrations of several serotype-specific antibodies among Indigenous adults may sound paradoxical. For interpretation of our data it is important to bear in mind that we assessed natural immunity in generally healthy adults. However, high burden of chronic diseases and immunocompromising conditions, along with unfavourable socioeconomic conditions, are important factors predisposing Indigenous people to invasive bacterial disease [40,72]. As Indigenous Canadians have high burden of obesity, diabetes, circulatory diseases, chronic kidney disease, and cancer, this suggests that an increased IPD rate in Indigenous adults is due to the increased prevalence of risk factors rather than lack of anti-capsular antibodies [73,74]. Indeed, according to our recent data, among northwestern Ontario adults with IPD, prevalence of immunocompromising conditions was much higher in Indigenous than in non-Indigenous patients (66% vs. 29.5%, p < 0.001) [21]. However, for interpretation of serological data, it is important to consider that anti-capsular antibodies detected by ELISA may have different functional abilities because of dissimilar avidity [75]. On the other hand, protection induced by the natural exposure to *S. pneumoniae* may depend on antibodies to non-capsular antigens, such as antibody to the pneumococcal cell wall polysaccharide or pneumococcal proteins [76,77].

Although serological correlates of protection against pneumococcal disease in adults have not been established, certain concentrations of specific IgG have been suggested for the interpretation of antibody responses to pneumococcal immunization. In the literature, the concentration of 1.3 μg/mL has been used as a cut-off for protection, and as an indicator of the adequate response to pneumococcal immunization in adults [78]; the concentration of 1 μg/mL was suggested as a threshold of protection against pneumococcal pneumonia in the elderly [79], although the protective threshold is serotype-dependent and may be different for naturally acquired as compared to post-vaccination antibodies. In our study, Indigenous individuals more frequently than non-Indigenous ones, had serotype-specific IgG above these thresholds, with statistically significant differences found for 6B, 9V, and 14, reflecting the differences in geometric mean antibody concentrations between the groups.

## Limitations and future directions

We analyzed natural antibodies specific to 7 select pneumococcal serotypes chosen for their diverse biological characteristics, significance in epidemiology, and inclusion in currently used vaccines. However, we did not study natural antibody specific to non-vaccine serotypes that

would be important considering the dynamic changes in epidemiology of pneumococcal infections and emergence of new serotypes in the post-conjugate vaccine era. In this study, we have reported only concentrations of serotype-specific IgG, IgA, and IgM antibodies. Although recognizing that analysis of the functional antibody abilities could provide better understanding of the protective capacities of natural antibodies, conducting the opsonophagocytic assays in our lab was not feasible.

There is always a potential for inaccurate information in any study when demographic data are collected by self-declaration. However, we made efforts to avoid recruiting participants with co-morbidities that could potentially impact their immunity, via asking them to complete a detailed questionnaire on their health status, with the list of common chronic conditions, co-morbidities, and history of taking corticosteroids or other immunosuppressive medications. As adult pneumococcal vaccination in Canada is not very common, especially among Indigenous populations, it would be very unlikely that they provided an incorrect record of pneumococcal vaccination.

A sample size was insufficient to complete analysis by age-stratified groups; non-Indigenous study participants from southern Ontario could not been recruited because of logistical reasons. Immunological correlates of protection against pneumococcal disease in adults have not been determined that presented challenges for clinical interpretation of collected data. The lack of data on pneumococcal carriage in the communities of interest precluded us from making direct conclusions on the effect of different carriage rates on the amounts of natural antibody in respective population groups. Collecting accurate data on carriage is problematic mainly due to its transitory manner that requires longitudinal studies and as it was previously recognized, "even with monthly collection of swabs, episodes of carriage are likely to be missed" [42]. For interpretation of our findings, we considered previously published data on pneumococcal carriage in different populations of various ages, while recognizing an inherent deficiency of such an approach. Finally, we recognize that Canadian Indigenous peoples represent a highly diverse group with different history, genetic backgrounds, cultural traditions, etc. While we made attempt to assess two different populations living in geographically diverse environments, it is impossible to generalize our findings towards other Indigenous groups.

## Conclusions

Comparison between healthy Indigenous and non-Indigenous adults suggests that increased rates of IPD and other forms of pneumococcal infections among Indigenous people are unlikely related to deficiency of naturally acquired antibodies, at least those specific to 7 common *S. pneumoniae* serotypes. Adverse social determinants of health, high burden of non-communicable chronic diseases, and increased prevalence of immunocompromising conditions most likely predispose Indigenous people to pneumococcal infections. One plausible solution to this problem would be the improvement of immunization coverage of adult groups at risk. Determining serological correlates of protection for adults will be essential to identify the groups in need of pneumococcal immunizations that may prevent an excessive burden of the disease among North American Indigenous people.

## Supporting information

**S1 Table. Participants with capsular polysaccharide specific IgG concentrations above 1.3 µg/ mL and 1.0 µg/ mL.** Fisher's exact test compared proportions of Indigenous and non-Indigenous participants with IgG concentrations > 1.3 µg/ mL and > 1.0 µg/ mL for each serotype.
(DOCX)

**S2 Table. Linear regression analysis of association of serotype-specific IgG antibodies with age for non-Indigenous groups (n = 64) and Indigenous (n = 76).** The goodness of the fit is reported by the $r^2$ value; the p value determines if the slope is significantly non-zero.
(DOCX)

**S3 Table. Linear regression analysis of association of serotype-specific IgA antibodies with age for non-Indigenous groups (n = 64) and Indigenous (n = 76).** The goodness of the fit is reported by the $r^2$ value; the p value determines if the slope is significantly non-zero.
(DOCX)

**S4 Table. Spearman correlation of IgM concentrations and age for non-Indigenous adults.**
(DOCX)

**S5 Table. Participant demographics based on ethnicity and sex.** Mann-Whitney U tests compared ages of males and females of the same ethnicity, there were no significant differences between groups. [a] One participant with an unknown age.
(DOCX)

**S6 Table. The geometric mean concentrations of anti-pneumococcal IgG, IgM and IgA antibodies (µg/ mL, 95% confidence intervals of the geometric mean) based on ethnicity and sex.** Concentrations of antibodies were compared between males and females of the same ethnicity using a Mann-Whitney U test.
(DOCX)

**S7 Table. Concentrations of naturally acquired serotype-specific pneumococcal IgG in healthy adults.**
(DOCX)

**S1 Appendix. Participant demographics.** Indigenous adults southern Ontario (Group 1), Indigenous adults northwestern Ontario (Group 2), non-Indigenous adults Thunder Bay (Group 3), and non-Indigenous adults Kenora (Group 4). Participant group, ID number, age and ethnicity are displayed.
(DOCX)

**S2 Appendix. Concentrations of pneumococcal capsular polysaccharide IgG (µg/ mL).** Indigenous adults southern Ontario (Group 1), Indigenous adults northwestern Ontario (Group 2), non-Indigenous adults Thunder Bay (Group 3), and non-Indigenous adults Kenora (Group 4). All data displayed are the original values. For our statistical analyses, the lower limit of detection was determined for each serotype according to the WHO pneumococcal ELISA protocol. The lower limits of detection are serotype 3 (0.100 µg/ mL), serotype 6B (0.167 µg/ mL), 9V (0.175 µg/ mL), 14 (0.339 µg/ mL), 19A (0.144 µg/ mL), 19F (0.166 µg/ mL) and 23F (0.144 µg/ mL). All values below the lower limits of detection were reported as half the value for statistical purposes.
(DOCX)

**S3 Appendix. Concentrations of pneumococcal capsular polysaccharide IgM (µg/ mL).** Indigenous adults southern Ontario (Group 1), Indigenous adults northwestern Ontario (Group 2), non-Indigenous adults Thunder Bay (Group 3), and non-Indigenous adults Kenora (Group 4). All data displayed are the original values. For our statistical analyses, the lower limit of detection was determined for each serotype according to the WHO pneumococcal ELISA protocol. The lower limits of detection are serotype 3 (0.035 µg/ mL), serotype 6B (0.088 µg/ mL), 9V (0.070 µg/ mL), 14 (0.124 µg/ mL), 19A (0.078 µg/ mL), 19F (0.130 µg/ mL) and 23F

(0.031 μg/ mL). All values below the lower limits of detection were reported as half the value for statistical purposes.
(DOCX)

**S4 Appendix. Concentrations of pneumococcal capsular polysaccharide IgA (μg/ mL).** Indigenous adults southern Ontario (Group 1), Indigenous adults northwestern Ontario (Group 2), non-Indigenous adults Thunder Bay (Group 3), and non-Indigenous adults Kenora (Group 4). All data displayed are the original values. For our statistical analyses, the lower limit of detection was determined for each serotype according to the WHO pneumococcal ELISA protocol. The lower limits of detection are serotype 3 (0.052 μg/ mL), serotype 6B (0.025 μg/ mL), 9V (0.036 μg/ mL), 14 (0.053 μg/ mL), 19A (0.025 μg/ mL), 19F (0.022 μg/ mL) and 23F (0.029 μg/ mL). All values below the lower limits of detection were reported as half the value for statistical purposes.
(DOCX)

**S5 Appendix. Questionnaire on Inclusivity in global research.**
(DOCX)

## Acknowledgments

We would like to thank all the participants who donated their serum for the study. We are indebted to our collaborators, especially the health care providers at Indigenous Health Access Centers and First Nations community members who kindly helped us with collecting samples, with our special thanks to Annette Schroeter, Connee Badiuk, Anita Cameron, Cynthia Price, Wayne Hyacinthe, Karen Fobister, Rekha Netto, Sherisse McLaughlin, Trudy Jacobs, and Lori Sinclair. We also thank Hanan Alsarmi for collecting the healthy Thunder Bay participant samples as well as Twyla Biluk (Thunder Bay 55 Plus Centre) for assistance with recruitment of healthy Thunder Bay participants. We thank Angele Desbiens-Forget, Brenda Huska (Northern Ontario School of Medicine) and Amanda Bakke for technical help.

## Author Contributions

**Conceptualization:** Marina Ulanova.

**Data curation:** Gabrielle N. Gaultier, Eli B. Nix, Joelle Thorgrimson.

**Formal analysis:** Gabrielle N. Gaultier.

**Funding acquisition:** Douglas Boreham, William McCready, Marina Ulanova.

**Investigation:** Gabrielle N. Gaultier, Eli B. Nix, Joelle Thorgrimson.

**Project administration:** William McCready, Marina Ulanova.

**Resources:** Douglas Boreham, Marina Ulanova.

**Supervision:** Marina Ulanova.

**Writing – original draft:** Gabrielle N. Gaultier.

**Writing – review & editing:** Marina Ulanova.

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
