## [Decision Letter · Decision Letter 0]

20 Jan 2022

PONE-D-21-38330Naturally acquired antibodies against 7 Streptococcus pneumoniae serotypes in Indigenous and non-Indigenous adultsPLOS ONE

Dear Dr. Ulanova,

Thank you for submitting your manuscript to PLOS ONE. After careful consideration, we feel that it has merit but does not fully meet PLOS ONE’s publication criteria as it currently stands. Therefore, we invite you to submit a revised version of the manuscript that addresses the points raised during the review process. During the revision process, please focus on addressing the comments related to study design, including the selection of serotypes, and data analysis and presentation.  Also, if there is a way to provide functional information regarding the antibody response it would be greatly appreciated.

We look forward to receiving your revised manuscript.

Kind regards,

Victor C Huber

Academic Editor

PLOS ONE

Journal Requirements:

"We would like to thank all the participants who donated their serum for the study. We are indebted to our collaborators, especially the health care providers at Indigenous Health Access Centers and First Nations community members who kindly helped us with collecting samples, with our special thanks to Annette Schroeter, Connee Badiuk, Anita Cameron, Cynthia Price, Wayne Hyacinthe, Karen Fobister, Rekha Netto, Sherisse McLaughlin, Trudy Jacobs, and Lori Sinclair.  We also thank Hanan Alsarmi for collecting the healthy Thunder Bay participant samples as well as Twyla Biluk (Thunder Bay 55 Plus Centre) for assistance with recruitment of healthy Thunder Bay participants. We thank Angele Desbiens-Forget, Brenda Huska (Northern Ontario School of Medicine) and Amanda Bakke for technical help.

Funding was provided by Mitacs Accelerate with contribution by Bruce Power, Northern Ontario Academic Medicine Association, Pfizer, and NOSM (Summer Medical Student Research Awards)."

We note that you have provided funding information. However, funding information should not appear in the Acknowledgments section or other areas of your manuscript. We will only publish funding information present in the Funding Statement section of the online submission form. 

"This work received support from: Mitacs Accelerate with contribution by Bruce Power; IT05441; www.mitacs.ca/en/programs/accelerate to MU, DB; Northern Ontario Academic Medicine Association (AHSC AFP Innovation Fund); www.noama.ca; A-12-06 to WM, MU; Pfizer (Investigator Initiated Research Project, Grant 53232197) to M.U. https://www.cybergrants.com/pfizer/Research; and NOSM Summer Medical Student Research Awards to EBN and JT, https://www.nosm.ca/research/student-research-at-nosm-2/deans-summer-medical-student-research-awards/. The funders had no role in study design, data collection and analysis, decision to publish, or preparation of the manuscript."

"I have read the journal's policy and the authors of this manuscript have the following

competing interests: Marina Ulanova received research funding from Pfizer through a

grant for Investigator Initiated Research Project to her institution, honoraria for serving on the advisory boards, and travel expenses from Pfizer. All other authors have declared that no competing interest exist."  

We note that you received funding from a commercial source: Pfizer

Within this Competing Interests Statement, please confirm that this does not alter your adherence to all PLOS ONE policies on sharing data and materials by including the following statement: ""This does not alter our adherence to PLOS ONE policies on sharing data and materials.” (as detailed online in our guide for authors http://journals.plos.org/plosone/s/competing-interests). If there are restrictions on sharing of data and/or materials, please state these. Please note that we cannot proceed with consideration of your article until this information has been declared. 

Reviewers' comments:

Reviewer's Responses to Questions

**Comments to the Author**

1. Is the manuscript technically sound, and do the data support the conclusions?

Reviewer #1: Yes

Reviewer #2: Yes

2. Has the statistical analysis been performed appropriately and rigorously? 

Reviewer #1: N/A

Reviewer #2: Yes

3. Have the authors made all data underlying the findings in their manuscript fully available?

Reviewer #1: Yes

Reviewer #2: Yes

4. Is the manuscript presented in an intelligible fashion and written in standard English?

Reviewer #1: Yes

Reviewer #2: Yes

5. Review Comments to the Author

Reviewer #1: This is an original and thought-provoking paper. Indigenous populations of non-vaccinated adults had higher levels of IgG and IgA to all pneumococcal serotypes tested than did non-Indigenous adults (Tables 2A and 2C). Antibody levels were unrelated to age.

The following specific suggestions might help with the clarity of the presentation.

What were the vaccination rates in infants and children in the Indigenous vs non-Indigenous populations at the time of the study?

The authors address well the principal question that the reader will, have, namely, why the antibody that is more prevalent in the population fails to provide protection:

“However, high burden of chronic diseases and immunocompromising conditions, along with unfavourable socioeconomic conditions, are important factors predisposing Indigenous people to invasive bacterial disease [35, 65]. As Indigenous Canadians have high burden of obesity, diabetes, circulatory diseases, chronic kidney disease, and cancer, this suggests that an increased IPD rate in Indigenous adults is due to the increased prevalence of risk factors rather than lack of natural immune defenses [66].”

These important points are summarized well in the conclusion.

I think, however, that better terminology in the preceding paragraph would be: “…rather than lack of anticapsular antibody.” There are other natural defenses, including those related to comorbidities, so this sentence should be rephrased. The authors also might add that some studies have shown that all antibody measured by ELISA is equally protective (Musher, Johnson and Watson Infect Immun 1990; Romero-Steiner et al Clin Infect Dis 1999).

Reviewer #2: This study by Gaultier and colleagues report the naturally-acquired pneumococcal serotype-specific antibody levels among Canadian indigenous adult populations compared with non-indigenous adults. They measured serum IgG, IgM and IgA to 7 serotypes contained in PCV13 and PPV23. This is a nicely written manuscript that adds to the evidence around the immune profiles among indigenous populations globally to try and understand the basis for their increased susceptibility to pneumococcal carriage and disease.

I have a few queries and suggestions for the authors to consider:

1. What was the rationale for selecting those 7 serotypes in particular? Only serotype 3 is a non-PCV13 type and is usually associated with lower responses. Have the authors considered extending the range of serotypes for examination to gain a better idea of the responses in these groups? Particularly since they state that 72% of IPD was caused by non-PCV13 types.

2. Many of the demographic information was obtained by self-declaration, including pneumococcal vaccination history and health status. How reliable do the authors think this is since we know the indigenous groups often have a higher proportion of co-morbidities that can effect immune responses?

3. In terms of the analysis, I think reporting GMCs is OK but for the IgG response, it would be good to also include the proportion >1.0ug/ml or >1.3ug/ml as this is what is often used to determine an adequate or protective response (even if not in the context of PPV23 vaccination). This might help with the interpretation of the data.

4. As the authors noted, the biological role of the IgA response observed needs further investigation. The values are very low so it would be helpful to have some discussion around this in terms of how this might provide protection.

5. The manuscript would greatly benefit from some analysis of functional antibody, whether by avidity or opsonophagcytosis since this would provide a better insight into whether these responses are truly protective among indigenous populations.

6. PLOS authors have the option to publish the peer review history of their article (what does this mean?). If published, this will include your full peer review and any attached files.

Reviewer #1: **Yes: **Daniel M. Musher MD, Distinguished Service Professor of Medicine, Baylor College of Medicine

Reviewer #2: No

---

## [Author Response · Author response to Decision Letter 0]

16 Mar 2022

1. We are re-submitting our revised manuscript following all the PLOS ONE’s style requirements, including those for file naming.

2. A complete copy of PLOS’ questionnaire on inclusivity in global research is included as S12 Appendix.

3. We have removed funding-related text from the manuscript. The current Funding Statement is correct.

4. Competing Interests Statement has been amended as follows.

“Marina Ulanova received research funding from Pfizer through a grant for Investigator Initiated Research Project to her institution, honoraria for serving on the advisory boards, and travel expenses from Pfizer. This does not alter our adherence to PLOS ONE policies on sharing data and materials. All other authors have declared that no competing interests exist.”

5. “Data not shown” is removed; the corresponding data are now provided as S2 and S3 tables.

6. Captions for Supporting Information files are now included at the end of the manuscript.

We have edited the manuscript according to the Reviewers' comments, in particular adding the rationale for selecting the serotypes, data on proportion of participants with IgG levels over specific thresholds, as well as specific data instead of “data not shown.” We have also made editorial changes according to the journal requirements. Several new references have been added to support the study design and conclusions; all minor suggestions and comments have also been addressed. All changes to the text have been clearly marked.

We hope that the manuscript will now be acceptable for publication in PLOS ONE. 

Please, see specific response to the Reviewers’ comments and suggestions below. 

Yours sincerely,

Marina Ulanova

Reviewer #1: This is an original and thought-provoking paper. Indigenous populations of non-vaccinated adults had higher levels of IgG and IgA to all pneumococcal serotypes tested than did non-Indigenous adults (Tables 2A and 2C). Antibody levels were unrelated to age.

The following specific suggestions might help with the clarity of the presentation.

What were the vaccination rates in infants and children in the Indigenous vs non-Indigenous populations at the time of the study?

Response: In many parts of Canada, including the province of Ontario, immunization coverage reports do not include any information of the race or ethnicity. Based on official data by Public Health Ontario, the average rates of immunization of school pupils in Ontario with pneumococcal conjugate vaccine during the time of our study were the following: 2015-2016: 79.0%, 2016-2017: 79.7%, 2017-2018: 74.1% (added to page 18, lines 360-364). Although from our personal communications with physicians practicing in Northern Ontario we learnt that both Indigenous and non-Indigenous families in the region generally have positive attitude to infant immunization, and vaccine hesitancy or denial are very uncommon, we cannot provide any formal data to support this. However, we included published data indicating that it was no difference between infant immunization coverage between the predominantly Inuit population in the North and the total population of the province of Quebec into the manuscript to provide some context (page 18, lines 364-368). 

The authors address well the principal question that the reader will, have, namely, why the antibody that is more prevalent in the population fails to provide protection:

“However, high burden of chronic diseases and immunocompromising conditions, along with unfavourable socioeconomic conditions, are important factors predisposing Indigenous people to invasive bacterial disease [35, 65]. As Indigenous Canadians have high burden of obesity, diabetes, circulatory diseases, chronic kidney disease, and cancer, this suggests that an increased IPD rate in Indigenous adults is due to the increased prevalence of risk factors rather than lack of natural immune defenses [66].”

These important points are summarized well in the conclusion.

I think, however, that better terminology in the preceding paragraph would be: “…rather than lack of anticapsular antibody.” There are other natural defenses, including those related to comorbidities, so this sentence should be rephrased. 

Response: We agree with the comment and have edited this sentence according to Dr. Musher’s suggestion (page 22, line 449).

The authors also might add that some studies have shown that all antibody measured by ELISA is equally protective (Musher, Johnson and Watson Infect Immun 1990; Romero-Steiner et al Clin Infect Dis 1999).

Response: We agree that it is important to emphasize that the protective activities of pneumococcal antibodies may vary and we have added several sentences with corresponding references to page 22 (lines 452-457).

Reviewer #2: This study by Gaultier and colleagues report the naturally-acquired pneumococcal serotype-specific antibody levels among Canadian indigenous adult populations compared with non-indigenous adults. They measured serum IgG, IgM and IgA to 7 serotypes contained in PCV13 and PPV23. This is a nicely written manuscript that adds to the evidence around the immune profiles among indigenous populations globally to try and understand the basis for their increased susceptibility to pneumococcal carriage and disease.

I have a few queries and suggestions for the authors to consider:

1. What was the rationale for selecting those 7 serotypes in particular? Only serotype 3 is a non-PCV13 type and is usually associated with lower responses. Have the authors considered extending the range of serotypes for examination to gain a better idea of the responses in these groups? Particularly since they state that 72% of IPD was caused by non-PCV13 types.

Response: We thank the Reviewer for this important comment. Their query provided us with an opportunity of extending Discussion to further address the great diversity of S. pneumoniae serotypes in terms of biological characteristics, immunogenicity, and abilities to cause invasive disease vs. asymptomatic carriage. To respond to this comment, we added several new sentences to pages 14-15 (lines 272-292). Of note, the serotype 3 is included in both PCV13 and PPV23, although, as the Reviewer correctly states, it indices a lower response to PCV13 immunization. However, we feel that longer discussion of peculiar immunogenic characteristics of serotype 3 would go beyond the scope of the present paper. This subject pertains mostly to issues of vaccine response. For example, it was suggested that immunological tolerance to serotype 3 capsular polysaccharide might develop following repeated vaccinations (as was discussed by De Wals, Commentary on paradoxical observations pertaining to the impact of the 13-valent pneumococcal conjugate vaccine on serotype 3 Streptococcus pneumoniae infections in children. Vaccine. 2018;36(37):5495-5496). 

To avoid unnecessary lengthy discussion (which would have been more appropriate for a review rather than an original paper) we decided to specifically focus on the points, which are most relevant to the study objectives. These are included on page 19 (lines 379-381 and 383-392).

Extending the range of serotypes for examination including non-vaccine serotypes is an excellent idea for a future study and we added this to Limitations and future directions (page 23, lines 471-475).

2. Many of the demographic information was obtained by self-declaration, including pneumococcal vaccination history and health status. How reliable do the authors think this is since we know the indigenous groups often have a higher proportion of co-morbidities that can effect immune responses?

Response: There is always a potential for inaccurate information in any study when demographic data are collected by self-declaration. However, we made efforts to avoid recruiting participants with co-morbidities that could potentially impact their immunity, via asking them to complete a detailed questionnaire on their health status, with the list of common chronic conditions, co-morbidities, and history of taking corticosteroids or other immunosuppressive medications. As adult pneumococcal vaccination in Canada is not very common, especially among Indigenous populations, it would be very unlikely that the participants provided an incorrect record of pneumococcal vaccination. This is now added to Limitations (page 23, lines 480-487).

3. In terms of the analysis, I think reporting GMCs is OK but for the IgG response, it would be good to also include the proportion >1.0ug/ml or >1.3ug/ml as this is what is often used to determine an adequate or protective response (even if not in the context of PPV23 vaccination). This might help with the interpretation of the data.

Response: Following this suggestion, we have added a supplementary table (S1 Table) with proportions of Indigenous and non-Indigenous study participants who had serotype-specific IgG concentrations >1.3 µg/ mL and >1 µg/ mL. The findings are described in Results (page 8, lines 166-168), and their interpretation added to Discussion (page 22, lines 458-468). 

4. As the authors noted, the biological role of the IgA response observed needs further investigation. The values are very low so it would be helpful to have some discussion around this in terms of how this might provide protection.

Response: We have added several sentences to the discussion on the biological role of serum IgA specific to S. pneumoniae (page 21, lines 433-440).

5. The manuscript would greatly benefit from some analysis of functional antibody, whether by avidity or opsonophagcytosis since this would provide a better insight into whether these responses are truly protective among indigenous populations.

Response: We agree that analysis of functional antibody activity would greatly add to understanding of their role in protection against pneumococcal disease. Unfortunately, conducting the opsonophagocytic assays in our lab was not feasible; this is now added to Limitations and directions of further study (page 23, lines 476-479).

---

## [Decision Letter · Decision Letter 1]

1 Apr 2022

Naturally acquired antibodies against 7 Streptococcus pneumoniae serotypes in Indigenous and non-Indigenous adults

PONE-D-21-38330R1

Dear Dr. Ulanova,

We’re pleased to inform you that your manuscript has been judged scientifically suitable for publication and will be formally accepted for publication once it meets all outstanding technical requirements.

Kind regards,

Victor C Huber

Academic Editor

PLOS ONE

Additional Editor Comments (optional):

Reviewers' comments:

Reviewer's Responses to Questions

**Comments to the Author**

1. If the authors have adequately addressed your comments raised in a previous round of review and you feel that this manuscript is now acceptable for publication, you may indicate that here to bypass the “Comments to the Author” section, enter your conflict of interest statement in the “Confidential to Editor” section, and submit your "Accept" recommendation.

Reviewer #2: All comments have been addressed

2. Is the manuscript technically sound, and do the data support the conclusions?

Reviewer #2: Yes

3. Has the statistical analysis been performed appropriately and rigorously? 

Reviewer #2: Yes

4. Have the authors made all data underlying the findings in their manuscript fully available?

Reviewer #2: Yes

5. Is the manuscript presented in an intelligible fashion and written in standard English?

Reviewer #2: Yes

6. Review Comments to the Author

Reviewer #2: The authors have addressed the comments sufficiently and the manuscript is acceptable for publication.

7. PLOS authors have the option to publish the peer review history of their article (what does this mean?). If published, this will include your full peer review and any attached files.

Reviewer #2: No

---

## [Editor Report · Acceptance letter]

5 Apr 2022

PONE-D-21-38330R1 

Naturally acquired antibodies against 7 *Streptococcus pneumoniae* serotypes in Indigenous and non-Indigenous adults 

Dear Dr. Ulanova:

I'm pleased to inform you that your manuscript has been deemed suitable for publication in PLOS ONE. Congratulations! Your manuscript is now with our production department. 

Kind regards, 

on behalf of

Dr. Victor C Huber 

Academic Editor

PLOS ONE